# Cavity Length Sequence Matching Algorithm Based on Combined Valley Peak Positioning for Fiber-Optic Fabry-Perot Sensors

**Liang Nie** [1,2], **Xiaonan Li** [1], **Hongwei Chen** [3], **Junying Zhang** [1,2], **Haibin Chen** [1,2,*], **Xueqing Zhao** [3], **Sheng Wang** [3] **and Guanghai Liu** [1,2]

1   School of Optoelectronics Engineering, Xi'an Technological University, Xi'an 710021, China; nieliang@xatu.edu.cn (L.N.); li_xiaonan_xatu@163.com (X.L.); zhangjunying@xatu.edu.cn (J.Z.); guanghailiu@163.com (G.L.)
2   Shaanxi Province Key Laboratory of Photoelectric Measurement and Instrument Technology, Xi'an Technological University, Xi'an 710021, China
3   State Key Laboratory of Laser Interaction with Matter, Northwest Institute of Nuclear Technology, Xi'an 710024, China; chenhongwei@nint.ac.cn (H.C.); zhaoxueqing@nint.ac.cn (X.Z.); wangsheng@nint.ac.cn (S.W.)
*   Correspondence: chenhaibin@xatu.edu.cn

**Abstract:** To solve the problem of low demodulation accuracy of conventional peak-to-peak algorithm for fiber-optic Fabry-Perot (FP) sensors due to failure of determining the interference order, a novel cavity length sequence matching demodulation algorithm based on a combined valley peak positioning is proposed. Firstly, a pair of a peak and its neighboring valley in the reflection spectrum is selected and positioned, and two groups of interference orders are supposed to generate two groups of cavity length sequences. Finally, these cavity lengths are compared to find the real interference order of the peak and valley for the extraction of the accurate cavity length. In order to verify the feasibility and performance of the proposed algorithm, simulations and experiments were carried out for fiber-optic FP sensors with cavity lengths in the range of 15–115 μm. A demodulation accuracy better than 8.8 nm was found. The proposed algorithm can achieve highly accurate cavity length demodulation of fiber-optic FP sensors.

**Keywords:** fiber-optic sensor; Fabry-Perot; peak-to-peak method; interference order

## 1. Introduction

Fiber-optic Fabry-Perot (FP) sensors have advantages of tiny size, strong environmental adaptability, and anti-electromagnetic interference, and can reliably monitor important parameters like pressure, temperature, strain, and so on, even in extreme environments. These sensors have been widely used in various fields such as aerospace, deep-sea exploration, bridge monitoring, petroleum drilling, biomedical and rehabilitation applications, etc. [1–8]. The measurement of environmental parameters is commonly achieved by the monitoring of the fiber-optic FP sensor's cavity length, therefore, the cavity length demodulation technology is one of the most important issues for real applications of fiber-optic FP sensors. The cavity length demodulation accuracy of the fiber-optic FP sensor directly affects the measurement accuracy.

Cavity length demodulation of fiber-optic FP sensors mainly includes the Fourier transform method [9,10], cross-correlation method [11,12], single-peak method [13,14], and peak-to-peak (P2P) method [15,16]. Among them, the Fourier transform and cross-correlation methods require multiple integral operations in the whole spectral range, resulting in slow demodulation speed and poor dynamic characteristics. The single-peak method can quickly extract the cavity length changes by tracking the characteristic peak of a certain interference order in the reflection spectrum, which has a high demodulation accuracy but

small measurement range because the absolute cavity length cannot be determined. The P2P method quickly calculates the cavity length by locating two characteristic peaks in the reflection spectrum, which has a wide demodulation range and a fast demodulation rate. However, in the demodulation process of the P2P method, the interference order of the peaks cannot be determined, which results in a low demodulation precision of the cavity length. The demodulation error of the P2P method is at least tens of times larger than that of the single-peak method.

To improve performance of the P2P method, several different kinds of improved algorithm were proposed. Jiang et al. proposed an improved P2P method [17] to calculate the cavity length by calculating the wavelength spacing of two adjacent spectral peaks in the spectrum through a linear fitting. The demodulation accuracy is improved about 25 times, but still on a 1 μm level. This method needs to locate multiple peaks in the spectrum to improve the fitting accuracy, and since the number of peaks in the spectrum will decrease with the decrease of cavity length, this method is not suitable for FP sensors with short cavity lengths. Chen et al. proposed a squared P2P algorithm, which can expand the demodulation range of the cavity length to short-cavity fiber-optic sensors, however, the demodulation error is still as large as 30 nm [18].

To improve cavity-length demodulation accuracy of fiber-optic FP sensors in the P2P method, this paper proposes a novel cavity-length sequence matching (CLSM) algorithm based on the combined positioning of a pair of neighboring peaks and valleys. From the comparison of the two cavity length sequences generated from the interference orders of the positioned neighboring peak and valley, the most matched interference order of the peak is found and the accurate cavity length is accordingly extracted. Both simulations and experiments are carried out to investigate the feasibility and performance of the proposed algorithm.

## 2. Principle of the CLSM Algorithm

### 2.1. Principle of Conventional P2P Algorithm

The typical structure of a fiber-optic FP sensor is shown in Figure 1a. Two vertically cut single-mode fibers (SMFs) are penetrated into a glass capillary (GC) with an inner diameter slightly larger than the outer diameter of the fiber cladding, and two parallel fiber ends form the FP sensor with a cavity length of $L$ [16,17]. A micrograph of the FP cavity formed by two fiber ends in a GC is given in Figure 1b.

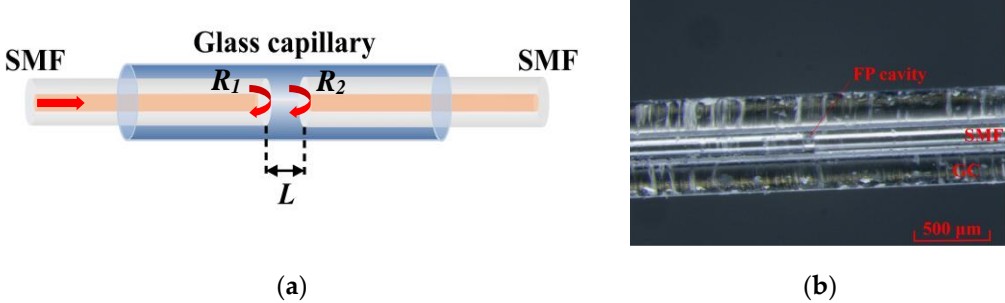

(**a**)                                       (**b**)

**Figure 1.** Typical structure of a fiber-optic FP sensor. (**a**) Schematic diagram, (**b**) micrograph of a FP sensor.

The reflectivity of the fiber-optic FP sensor can be give by

$$R_{FP}(\lambda) = \frac{R_1 + R_2 + 2\sqrt{R_1 R_2} \cos\left(\frac{4\pi n L}{\lambda} + \pi\right)}{1 + R_1 R_2 + 2\sqrt{R_1 R_2} \cos\left(\frac{4\pi n L}{\lambda} + \pi\right)}, \tag{1}$$

where $\lambda$ is the wavelength, $n$ is the refractive index of the FP cavity, $R_1$ and $R_2$ are the reflectivity of the two reflecting surfaces. For an air-gap FP sensor, $n = 1$.

If the fiber-optic FP sensor is illuminated by a broadband light source, the reflection spectrum will have a waveform with multiple peaks and valleys. For an air-gap FP sensor, the spectral peaks satisfy

$$\frac{4\pi L}{\lambda_m} + \pi = 2\pi m, \ m = 1, \ 2, \ 3\ldots, \tag{2}$$

where $m$ is the interference order.

Considering two different interference orders m and $m + q$, where q is the difference between the two interference orders, we have

$$L = \frac{q}{2}\left(\frac{\lambda_m \lambda_{m+q}}{\lambda_m - \lambda_{m+q}}\right)q, m = 1, 2, \ 3\ldots, \tag{3}$$

From the wavelength positioning of two different spectral peaks, the cavity length of the fiber-optic FP sensor can be extracted by Equation (3), which is exactly the core principle of the conventional P2P algorithm.

It can be deduced from Equation (3) that when two neighboring peaks are selected, the cavity length demodulation error of the P2P method is

$$\Delta d = \sqrt{2}md\left|\frac{\Delta\lambda_m}{\lambda_m}\right| = 2\sqrt{2}\left(\frac{d}{\lambda_m}\right)^2 \Delta\lambda_m \qquad m = 1, \ 2, \ 3\ldots, \tag{4}$$

where $\Delta\lambda_m$ is the positioning error of the $m$-th order peak. According to Equation (4), the measurement error of the P2P algorithm is directly proportional to the square of the ratio of the cavity length and peak wavelength. When the cavity length $L = 100$ μm and the central wavelength is 1568 nm. Even if the peak positioning error is in a picometer level ($\Delta\lambda_m = 30$ pm), the measurement error reaches a large value of $\Delta d = 0.345$ μm. The main reason is that the P2P algorithm can not determine the interference order of the characteristic peak in the process of the cavity length demodulation, which results in a low demodulation accuracy.

### 2.2. Principle of CLSM Algorithm

To solve the problem that the interference order cannot be determined in the demodulation process of the P2P method, we propose a CLSM algorithm based on a combined valley and peak positioning process. The principle for the new algorithm is discussed below.

Convert the reflection spectrum into the frequency domain by $\lambda = c/v$ ($v$ is the optical frequency and $c$ is the vacuum speed of light) into Equation (2), we have

$$R_{FP}(\lambda) = \frac{R_1 + R_2 + 2\sqrt{R_1 R_2}\cos\left(\frac{4\pi L}{c}v + \pi\right)}{1 + R_1 R_2 + 2\sqrt{R_1 R_2}\cos\left(\frac{4\pi L}{c}v + \pi\right)}. \tag{5}$$

It can be seen that the reflectivity is a periodic function of the frequency in the frequency domain, and the period can be expressed as

$$T = \frac{2\pi}{4\pi L/c} = \frac{c}{2L}. \tag{6}$$

In the frequency domain, select a pair of adjacent peaks and valleys. If the interference order of the selected peak is $m_p$, the corresponding peak frequency is $v_p$, and $m_p$ is also the total number of peaks in the optical frequency range from 0 to $v_p$. The period of the spectral signal in the frequency domain can also be expressed as

$$T = \frac{v_p}{m_p}. \tag{7}$$

From Equations (6) and (7), the cavity length of the single peak demodulation FP sensor is

$$L = \frac{cm_p}{2v_p}. \tag{8}$$

Because the interference order of the selected peak is unknown, the cavity length cannot be calculated directly. In order to accurately judge the interference order of the selected peak, a group of interference order sequences $\{N\}$, $N = 1, 2, 3 \ldots$ are introduced, in which it is required that $N$ covers the interference order $m_p$. For each $N$, two groups of cavity length sequences $\{d_{Np}\}$ and $\{d_{Npp}\}$ can be generated by

$$d_{Np} = \frac{cN}{2v_p}, \ N = 1, 2, 3 \ldots, \tag{9}$$

and

$$d_{Npp} = \frac{c(N - 0.5)}{2v_{pp}}, \ N = 1, 2, 3 \ldots. \tag{10}$$

where $v_p$ and $v_{pp}$ are the center frequencies corresponding to the selected adjacent peak and valley respectively, $d_{Np}$ and $d_{Npp}$ are the generated cavity lengths corresponding to $v_p$ and $v_{pp}$ for a supposed interference order of $N$. If $N$ is not the real interference order of the selected adjacent peak and valley, the two generated cavity lengths deviate from each other, i.e., $d_{Np} \neq d_{Npp}$. If and only if $N$ equals to the real interference order, i.e., $N = m_p$, then, $d_{Np} = d_{Npp}$, which is also the real cavity length. Thus, by the continuous comparison of the generated cavity length pair of each interference order, the real interference order and real cavity length can be determined.

Considering that the existence of a peak positioning error always makes the central frequencies of the peak and valley pair cannot be accurately determined, even when the real interference order is found, there still exists a deviation between the two generated cavity lengths. We can calculate the deviations between each pair of the generated cavity lengths by $\{\Delta d_N\}$, $\Delta d_N = |d_{Np} - d_{Npp}|$, $N = 1, 2, 3 \ldots$ and search the minimum value of $\Delta d_N$.

When $\Delta d_N$ takes the minimum value, $N = m_p$, the real interference order is then determined, and the cavity length corresponding to the interference order is closest to the real cavity length. The finally determined cavity length can be given by the average of the two generated cavity lengths corresponding to the minimum deviation, which can be expressed as

$$d = \frac{d_{Np} + d_{Npp}}{2}. \tag{11}$$

Taking $\lambda = c/v$ into Equations (9) or (10), the cavity length measurement error of the CLSM algorithm caused by the peak or valley positioning can be given by

$$\Delta d = \frac{d}{\lambda} \Delta \lambda. \tag{12}$$

It can be seen from the above formula that the measurement error $\Delta d$ of the demodulation cavity length of the CLSM algorithm is directly proportional to the ratio of the cavity length and peak wavelength., while the measurement error of the traditional P2P method is directly proportional to the square of the ratio of the cavity length and peak wavelength. Compared to the traditional P2P method, the measurement error of the CLSM algorithm can be greatly reduced. When the cavity length $d = 100$ μm, the selected peak center wavelength $\lambda = 1568$ nm, considering a peak positioning error $\Delta \lambda_m = 30$ pm, the measurement error $\Delta d$ of the CLSM algorithm is only 2 nm.

## 3. CLSM Algorithm and Simulation Analysis

### 3.1. CLSM Algorithm

According to the demodulation principle discussed above, a CLSM algorithm is proposed for the cavity length extraction of fiber-optic FP sensors. The flowchart is shown in Figure 2, and the specific procedures are as below.

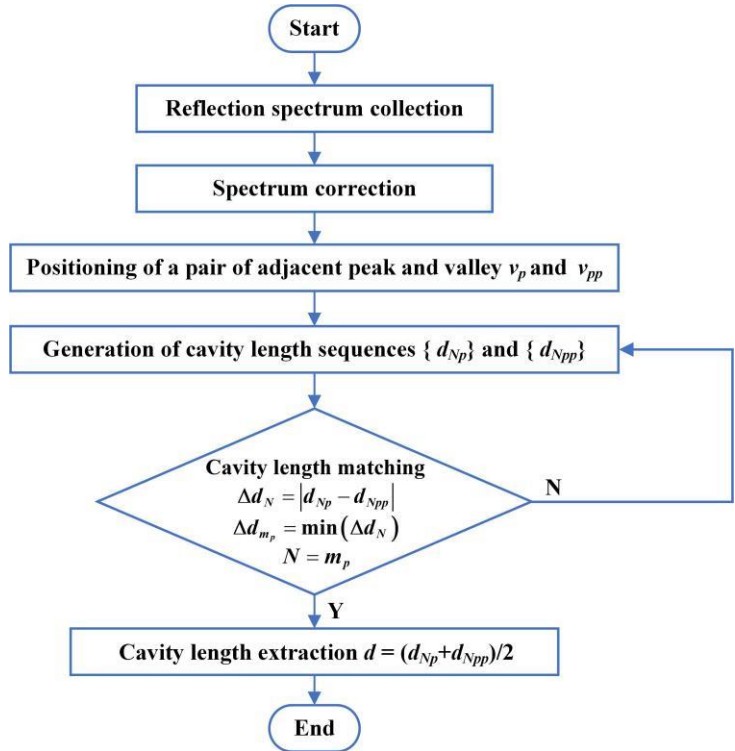

**Figure 2.** Flowchart of the CLSM algorithm.

First, collect the reflection spectrum of the fiber-optic FP sensor by a digital optical spectrum analyzer (OSA).

Second, to remove the non-uniformity distribution of the spectrum that comes from the light source, the reflection spectrum acquired by the OSA is divided by the optical spectrum of the light source itself. By this operation, the reflection spectrum will be corrected to have the same amplitude.

Third, the optical spectrum in the wavelength domain is transformed into the frequency domain, and a pair of adjacent peaks and valleys are selected, and their central frequencies $v_p$ and $v_{pp}$ are precisely positioned.

Fourth, two groups of interference order sequences are respectively generated according to Equations (9) and (10) by taking into positioned frequencies of the adjacent peak and valley. Accordingly, two groups of cavity length sequences $\{d_{Np}\}$ and $\{d_{Npp}\}$ are obtained.

Finally, the two groups of cavity length sequences are compared. By calculating the deviations $\{\Delta d_N\}$, $\Delta d_N = |d_{Np} - d_{Npp}|$, $N = 1, 2, 3 \ldots$ between each pair of the generated cavity lengths, the most-matched two cavity lengths are determined by searching the minimum difference value $\min(\Delta d_N)$ of the cavity lengths coming from the two sequences, and the demodulated cavity length is extracted by computing the average of the most-matched two cavity lengths according to Equation (11).

### 3.2. Simulation Analysis

To verify the feasibility of the proposed CLSM algorithm, we simulately gave out the reflection spectra of several fiber-optic FP sensors with cavity lengths in the range of

15–115 μm; then the reflection spectra were processed by a program written according to the proposed CLSM algorithm.

For a fiber-optic FP sensor with the structure shown in Figure 1, assuming its cavity length is 60 μm and it is illuminated by an SLD with a Gaussian spectrum, the reflection spectrum in a wavelength range of 1440–1675 nm can be given as Figure 3a. The spectrum in the frequency domain after spectrum correction is shown in Figure 3b, in which a pair of adjacent peaks and valleys are selected. After introducing interference order sequences, two groups of corresponding cavity length sequences were generated, as given in Figure 3c. The cavity length differences between the same order of the cavity length sequences were also given. It can be seen that, at the interference order $N = 74$, the difference between the two cavity length sequences reached the minimum, thus, this interference order can be determined to be the real order, the peak and valley corresponding cavity lengths were 59.9994 μm and 59.9997 μm, respectively. According to the CLSM algorithm, the demodulated cavity length can be extracted as the average of the two values, and the final result was 59.9995 μm, which deviates from the predetermined value 60 μm only 0.5 nm. Thus, the demodulation accuracy of conventional P2P method can be significantly improved through the proposed CLSM algorithm.

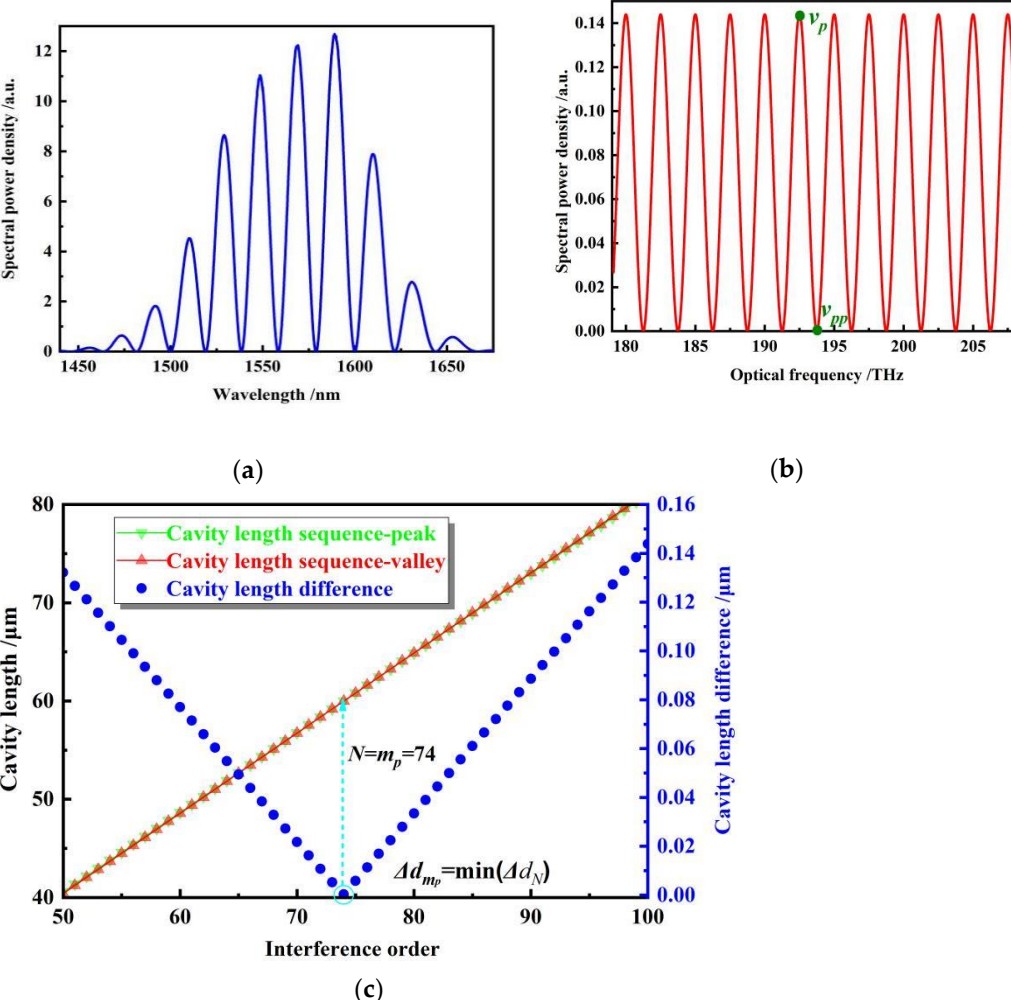

**Figure 3.** Simulating demodulation of a 60-μm fiber-optic FP sensor through the CLSM algorithm. (**a**) Reflection spectrum; (**b**) corrected reflection spectrum in the frequency domain; (**c**) generated cavity length sequences and their differences.

Demodulation performances of the CLSM algorithm are further simulatedly investigated by the processing of other fiber-optic FP sensors with cavity lengths in the range

of 10–120 μm; the simulation results are shown in Table 1 and Figure 4. It can be found that the maximum demodulation error of the CLSM algorithm was about 1.1 nm, in comparison, the maximum demodulation error of the conventional P2P algorithm was about 39.7 nm. It can be confirmed that the proposed CLSM algorithm can effectively improve the demodulation accuracy.

**Table 1.** Simulated demodulation results for 10–120 μm FP cavities through the CLSM and double-peak algorithm.

| Predetermined Cavity Length (μm) | Demodulated Cavity Length (μm) | | Demodulation Error (nm) | |
|---|---|---|---|---|
| | CLSM | P2P | CLSM | P2P |
| 10.0000 | 9.9999 | 10.0010 | 0.1 | 1.0 |
| 20.0000 | 19.9998 | 20.0033 | 0.2 | 3.3 |
| 30.0000 | 29.9998 | 30.0049 | 0.2 | 4.9 |
| 40.0000 | 39.9997 | 40.0108 | 0.3 | 10.8 |
| 50.0000 | 49.9996 | 50.0267 | 0.4 | 26.7 |
| 60.0000 | 59.9995 | 60.0236 | 0.5 | 23.6 |
| 70.0000 | 69.9995 | 69.9754 | 0.5 | 24.6 |
| 80.0000 | 79.9993 | 79.9645 | 0.7 | 35.5 |
| 90.0000 | 89.9992 | 90.0382 | 0.8 | 38.2 |
| 100.0000 | 99.9991 | 100.0385 | 0.9 | 38.5 |
| 110.0000 | 109.9991 | 110.0383 | 0.9 | 38.3 |
| 120.0000 | 119.9989 | 119.9603 | 1.1 | 39.7 |

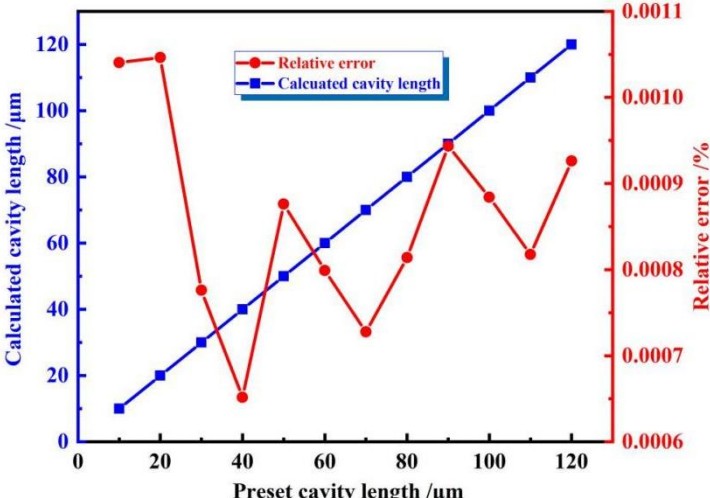

**Figure 4.** Relationships of the simulated cavity length and the error versus the preset cavity length.

The relationships of the simulatedly demodulated cavity length and the demodulation error versus the preset cavity length are shown in Figure 4. It can be seen that there was a good consistency between the calculated results and the preset cavity length. An R2 coefficient of 0.9999 can be obtained by linear fitting, and the relative error is at most 0.001%. Thus, the feasibility and good performances of the CLSM algorithm were successfully verified by the simulation.

## 4. Experimental Verification

To verify practical performances of the CLSM algorithm, an experimental setup for the cavity length demodulation of fiber-optic FP sensors was built, as shown in Figure 5. The system consists of a superluminescent diode (SLD, 3-dB spectral range: 1523~1613 nm), an optical circulator, and an optical spectrum analyzer (OSA, Antitsu, Japan, model: MS9740A, best resolution: 30 pm). The fiber-optic FP sensor can be directly connected to the system through port 2 of the optical circulator.

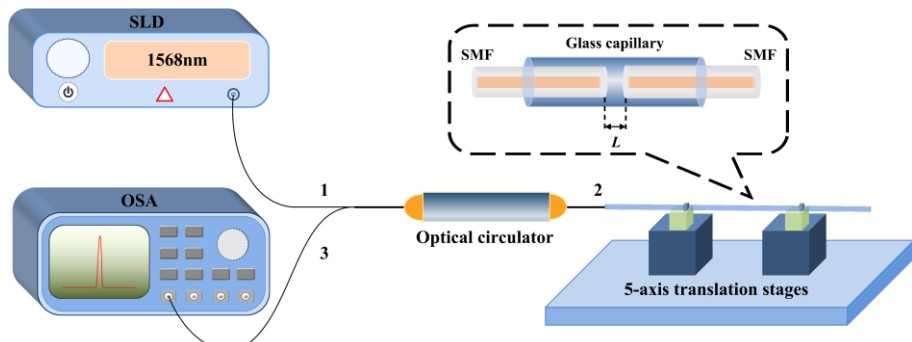

**Figure 5.** Experimental system for the cavity length demodulation of fiber-optic FP sensors.

The wideband light emitted from the SLD is inputted into port 1 of the optical circulator, and is outputted and illuminated on the fiber-optic FP sensor through port 2. Part of the light is reflected back, which carries on the modulation information from the FP sensor. The reflected light passes through port 2 of the optical circulator and is outputted by its port 3, and finally, received by the OSA to obtain the reflection spectrum. By processing the reflection spectrum on a computer through a program written according to the proposed CLSM algorithm, the cavity length of the fiber-optic FP sensor is extracted.

The tested fiber-optic FP sensors are fabricated according to the structure shown in Figure 1. Nine FP sensors with different cavity lengths of 16.4698 μm, 29.3783 μm, 41.0298 μm, 53.2906 μm, 64.1130 μm, 76.3849 μm, 88.1071 μm, 102.0809 μm, and 113.5901 μm were fabricated and connected to the experimental demodulation system one by one.

In Figure 6a, the reflection spectrum of a 64.1130 μm fiber-optic FP sensor is given. After amplitude correction and wavelength-to-frequency transformation, the corrected spectrum is shown in Figure 6b. A pair of adjacent peaks and valleys is selected, and the corresponding center frequencies are $v_p = 196.5094$ THz and $v_{pp} = 195.3397$ THz. Then the corresponding cavity length sequences are generated according to Equations (9) and (10), as shown in Figure 6c. The cavity length differences between each pair of the cavity lengths with the same interference order are also given. It can be observed that when the interference order was 84, the cavity length difference had the minimum value, correspondingly, the cavity lengths generated ware 64.1191 μm and 64.1191 μm, respectively, and the average cavity length was 64.1191 μm, which deviates from the real cavity length about 6.1 nm, thus, the error was only 6.1 nm.

Demodulated cavity lengths and errors of the nine fiber-optic FP sensors with different cavity lengths are shown in Table 2 and Figure 7, where all the results were gotten from one-time sampling of the spectrum without any averaging. It was found that the maximum error of the CLSM algorithm was 8.8 nm. In comparison, the maximum demodulation error of the traditional P2P algorithm was 5.5 μm. Thus, in the real experimental demodulation, the demodulation error of the CLSM algorithm was also much improved compared to the traditional P2P method.

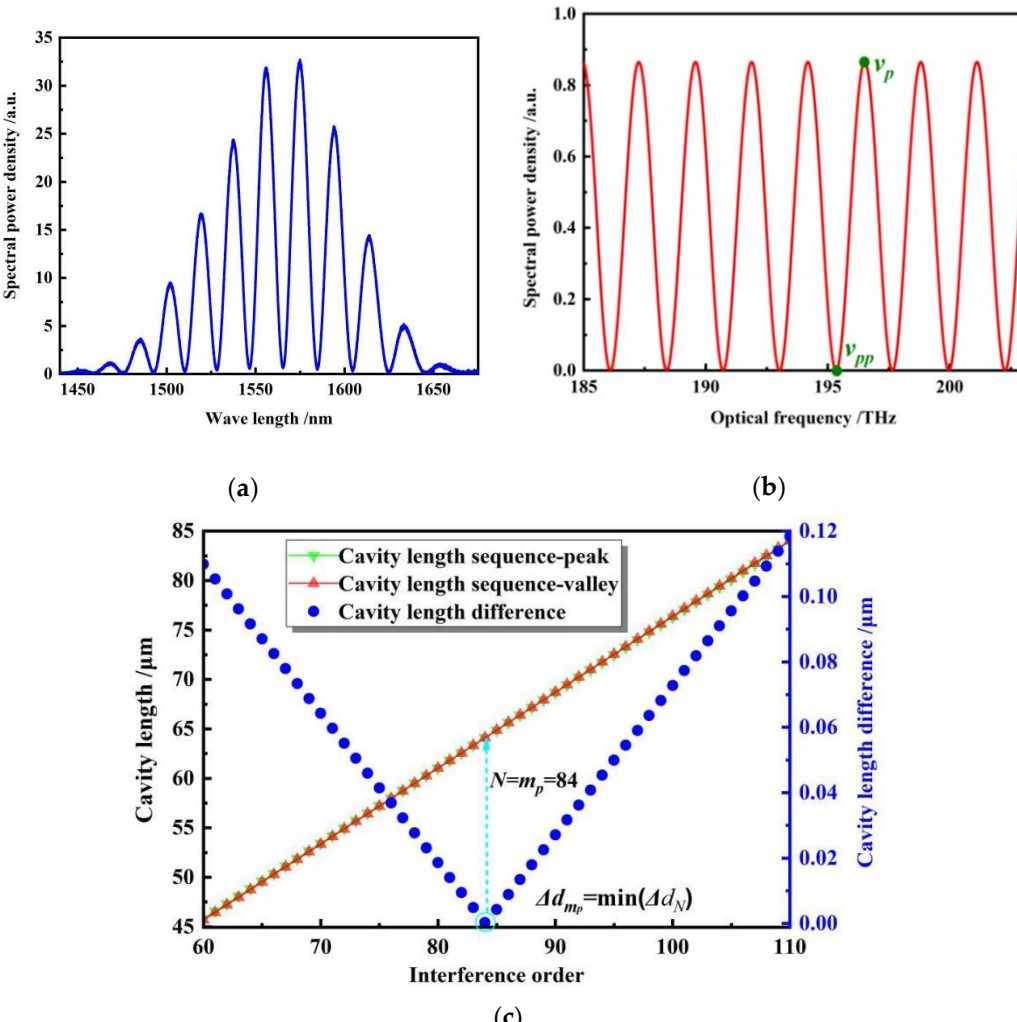

(a)

(b)

(c)

**Figure 6.** Spectra and cavity length sequences for a 64.1130-μm fiber-optic FP sensor. (**a**) Reflection spectrum; (**b**) corrected spectrum in the frequency domain; (**c**) generated cavity length sequences and their differences.

**Table 2.** Experimental demodulation results of 15–115-μm fiber-optic FP sensors by the CLSM algorithm.

| Predetermined Cavity Length (μm) | Demodulated Cavity Length | | Demodulation Error | |
|---|---|---|---|---|
| | CLSM (μm) | P2P (μm) | CLSM(nm) | P2P(μm) |
| 16.4698 | 16.4656 | 18.7605 | 4.2 | 2.3 |
| 29.3783 | 29.3735 | 32.0028 | 4.8 | 2.6 |
| 41.0298 | 41.0245 | 43.8287 | 5.3 | 2.8 |
| 53.2906 | 53.2961 | 56.8075 | 5.5 | 3.5 |
| 64.1130 | 64.1191 | 68.1084 | 6.1 | 4.0 |
| 76.3849 | 76.3916 | 80.9057 | 6.7 | 4.5 |
| 88.1071 | 88.0998 | 92.9886 | 7.3 | 4.9 |
| 102.0809 | 102.0893 | 107.3067 | 8.4 | 5.2 |
| 113.5901 | 113.5813 | 108.0648 | 8.8 | 5.5 |

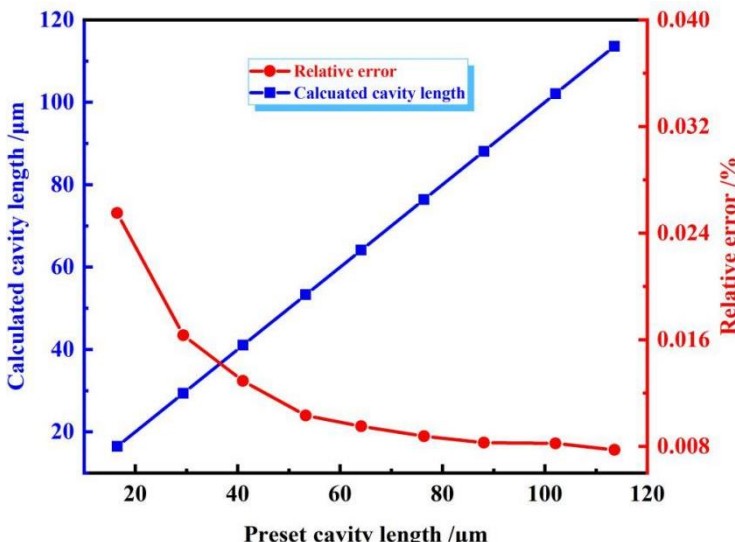

**Figure 7.** Relationships of the demodulated cavity length (**left**) and demodulation error (**right**) versus the standard cavity length.

Finally, to give out an evaluation of the demodulation resolution of the proposed CLSM algorithm, FP sensors with four different representative cavity lengths of 16.4698, 41.0298, 64.1130, and 88.1071 μm were continuously demodulated 100 times. As an example, the demodulated results for the cavity length of 64.1130 are shown in Figure 8. It was found that the maximum value of the cavity length was 64.1199 μm, the minimum value was 64.1181 μm, the average value was 64.1190 μm, which deviated from the preset cavity length of about 0.0059 μm, the variation range was 1.8 nm, and the standard deviation was 0.45 nm. Thus, an accuracy of 6 nm and a resolution of 0.45 nm were achieved. For the other three cavity lengths of 16.4698, 41.0298, and 88.1071 μm, through a similar way, the resolution was evaluated to be 0.17, 0.27, and 0.97 nm, respectively. Obviously, the resolution was gradually decreased with the increase of the cavity length, however, the relative results preserved a much better level.

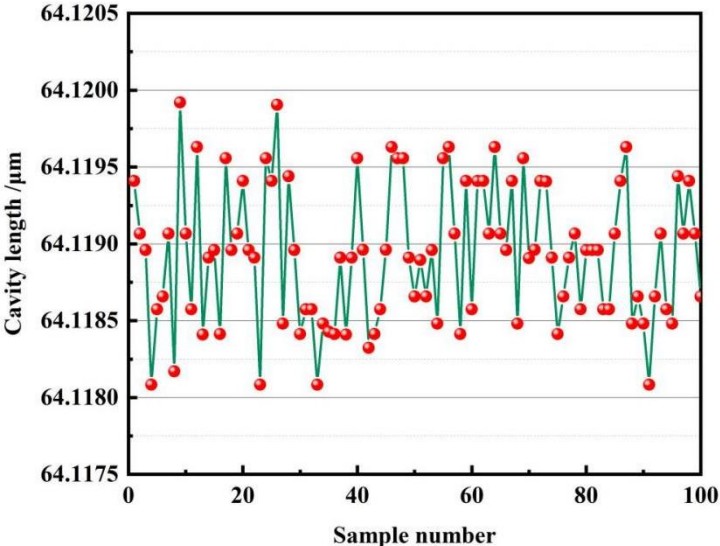

**Figure 8.** Repeated measurement of a fiber-optic FP sensors for 100 times.

## 5. Conclusions

In conclusion, a novel cavity length sequence matching algorithm based on a combined valley peak positioning approach for the accurate demodulation of fiber-optic FP sensors

is successfully proposed and demonstrated. A pair of neighboring peaks and valleys is positioned to generate two cavity length sequences, which are compared to get the most matched interference order, and from this interference order, the accurate cavity length is extracted. The algorithm solves the problem of low demodulation accuracy of fiber-optic FP sensors of the conventional P2P demodulation method due to the failure to determine the interference order. Simulations and experiments for fiber-optic FP sensors with cavity lengths in a range of 15–115 μm show a maximum error of 8.8 nm. The proposed algorithm can significantly improve the demodulation accuracy relative to the conventional P2P algorithm, which may be used in practical applications of fiber-optic FP sensors. Furthermore, with the update of a few parameters, the proposed algorithm can also be used for the accurate demodulation of other interferometric-type fiber-optic sensors, such as Mach–Zenhder interferometric sensors, through the optical path difference calculation.

**Author Contributions:** Conceptualization, L.N. and H.C. (Haibin Chen); methodology, H.C. (Hongwei Chen); software, X.L. and G.L.; validation, J.Z.; formal analysis, J.Z.; investigation, S.W.; writing—original draft preparation, L.N. and X.L.; writing—review and editing, H.C. (Haibin Chen) and L.N.; project administration, X.Z.; funding acquisition, H.C. (Haibin Chen). All authors have read and agreed to the published version of the manuscript.

**Funding:** This research was funded by the Natural Science Basic Research Project of Shaanxi Province, China, grant number 2021JM-437, and the Foundation of State Key Laboratory of Laser Interaction with Matter, grant number SKLLIM2007.

**Institutional Review Board Statement:** Not applicable.

**Informed Consent Statement:** Not applicable.

**Data Availability Statement:** Not applicable.

**Conflicts of Interest:** The authors declare no conflict of interest.

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
