# Peer review of "Cavity Length Sequence Matching Algorithm Based on Combined Valley Peak Positioning for Fiber-Optic Fabry-Perot Sensors"

_photonics, doi:10.3390/photonics9070451_

Round 1
Reviewer 1 Report
This paper reports a cavity length sequence matching algorithm based on combined valley peak positioning for fiber-optic FP sensors. I have some comments.
1. Introduction must be improved in terms of novelty of FP sensors for many other applications to highlight the advantages, when is claimed "These sensors have been widely used in various fields such as aerospace, deep sea exploration, bridge monitoring, petroleum drilling, etc []." I miss some references about FP usage for interrogators fabrication and "biomedical, rehabilitation applications" and add these words in the phrase and include the following references: IEEE Sensors Journal 19 (21), 9798-9805, 2021; Measurement 124, 486-493, 2018.
2. Fig. 6. How can guarantee similar spectra for different probes, with identical performance? How complex can be the fabrication of such and how can overcome it to produce reproducibility?
3. Table 2: how many tests were done for each Predetermined cavity length? can we get repeatability for the same length for all lengths? Fig. 8 show for one length, can we have a statistical value for other? Please be clear.
4. Can this algorithm to be used for other MZI sensors? Please comment.
Author Response
Thanks for careful reading and the constructive comments. The comments are valuable and very helpful for revising and improving our paper. We have accordingly made necessary modifications, which were listed in a point-by-point style. Please see the attachment.

Reviewer 2 Report
In this paper, the authors present one method to solve the problem of low demodulation accuracy of the conventional peak-to-peak algorithm for fiber-optic Fabry-Perot (FP) sensors due to failure of determining the interference order, a novel cavity length sequence matching demodulation algorithm based on a combined valley peak positioning. A demodulation accuracy better than 8.8 nm was found. The proposed algorithm can achieve highly accurate cavity length demodulation of fiber-optic FP sensors. This article is clear, concise, and suitable for the scope of the journal. Only several small suggestions are supplied:
1. Suggest the authors' supply one picture of the FP cavity with a length label.
2. Suggest the authors rewrite the CLSM algorithm part with more detail.
3. Suggest the authors enhance the introduction part with some latest reviews such as:
For deep-sea exploration:
Optical fiber sensing for marine environment and marine structural health monitoring: A review Optics and Laser Technology, 2021.
etc.
Author Response
Thanks for careful reading and the constructive comments. The comments are valuable and very helpful for revising and improving our paper. We have accordingly made necessary modifications, which were listed in a point-by-point style. Please see the attachment for the detailed response.

Round 2
Reviewer 1 Report
Satisfatory revision. Thanks